# Comparison of Distance and Angular Analysis for Measurement of Hamstring Flexibility in Preschoolers

**DOI:** 10.3390/children9010039

**Published:** 2022-01-02

**Authors:** Anke Hua, Jingyuan Bai, Yong Fan, Jian Wang

**Affiliations:** 1Department of Sports Science, College of Education, Zhejiang University, Hangzhou 310058, China; hak@zju.edu.cn (A.H.); baijy@zju.edu.cn (J.B.); 2Department of Sports, Qianjiang College, Hangzhou Normal University, Hangzhou 311121, China; fanyong@huqc.edu.cn; 3Center for Psychological Sciences, Zhejiang University, Hangzhou 310058, China

**Keywords:** hamstring flexibility, age-related change, sit-and-reach

## Abstract

The study aimed to (1) investigate the reliability and usefulness of a proposed angular analysis during a modified sit-and-reach (MSR) test, and (2) compare the proposed MSR angular analysis and the commonly used MSR distance to verify the influence of the anthropometric characteristics in preschoolers. 194 preschoolers participated in the study. Before testing, the anthropometric characteristics were collected. Each participant performed the MSR test twice. The MSR distance score was obtained from the starting point to the reaching point, while the MSR angle score was calculated according to the approximate hip flexion angle. Both the relative and absolute reliability were good for the angular analysis during an MSR test in preschoolers (ICC ranging from 0.82 to 0.91, CV% ranging from 8.21 to 9.40). The angular analysis demonstrated good usefulness, with a lower typical error than the smallest worthwhile change in 3- and 5-year-old groups. The MSR angle scores could eliminate the concern of the influence of anthropometric characteristics, while MSR distance and anthropometric characteristics (i.e., sitting height and arm length) were found to be weakly correlated. In conclusion, the angular analysis when performing the MSR test is reliable and appears to eliminate the concern regarding the limb length bias.

## 1. Introduction

As a major health-related fitness quality, flexibility (i.e., maximum joint range of motion) has been included in many fitness test batteries. Good flexibility might be helpful in enhancing performance and reducing the risk of sports injuries. The sit-and-reach (SR) test is the most common field test used to measure hamstring and low back flexibility [1]. The classical SR test was designed by Wells and Dillon [2], which has a simple procedure and is easy to administer. The most common assumption when interpreting the SR scores is that participants with better scores have a better ability of trunk and hamstring flexibility than those with lower scores [3]. The SR score is always a positive number, even for children who are unable to reach their toes [4].

However, previous studies proposed that the classical SR test did not allow for proportional differences between arm and leg lengths since they found that individuals with high finger-to-box distance measurements demonstrated a lower score on the SR test [5]. Therefore, these researchers proposed a modified sit-and-reach (MSR) test and established a relative zero point (i.e., the finger-to-box distance when the head and back in contact with the wall). The MSR test scores are measured from the relative zero point to the final reaching point and appear to eliminate the concern of disproportionate limb-length bias [6].

Although the MSR test does help control for arm-leg length discrepancies, Castro-Piñero et al. [7] suggested that the MSR test is not a more valid method to assess low back and hamstring flexibility than the SR test in children and adolescents since the criterion-related validity of the SR and MSR test was both weak. Besides, the effects of limb length differences on MSR test scores should also be considered since Hemmatinezhad et al. [8] found that the correlations between MSR scores with anthropometric characteristics (i.e., sitting height, arm length, leg length, trunk length, arm span and sum of trunk and arm lengths) were all significant (r ranging from 0.23–0.38, *p* < 0.05). Particularly, the results from Hemmatinezhad et al. [8] suggested that arm span had the strongest correlation with MSR reaching scores. Both the SR and MSR test measure the final position that the participants reach, but not the angle of the hip joint. Thus, angular tests were proposed for calculating the hip joint angle using an inclinometer or the angular kinematic analysis, respectively [4,9], since the angle scores are not influenced by anthropometric factors or spinal mobility.

Theoretically, hamstring flexibility is characterized by the maximum range of motion in a joint or series of joint [10]. Nevertheless, due to the qualified technicians, additional devices and time constraints, the use of the angular tests seems to be limited in several settings, including a school context.

Thus, our present study proposes a method for calculating the approximate hip flexion angle when performing the MSR test without additional equipment. The purpose of this study was to (a) propose a method for calculating the MSR angle to evaluate the hamstring flexibility and investigate its reliability and usefulness characteristics, and (b) determine whether the MSR angle scores can negate the effects of anthropometric characteristics, and (c) evaluate the hamstring flexibility in preschoolers using two MSR scores. We hypothesized that (a) the newly-proposed MSR angular analysis would be reliable and useful, and (b) MSR angular scores would eliminate the concern regarding the limb length bias while MSR distance scores would be correlated with anthropometric characteristics. We also hypothesized that preschool-aged girls would have better flexibility than boys.

## 2. Materials and Methods

### 2.1. Subject

A total of 194 (119 boys and 75 girls) healthy children aged 3–6 years participated in the study. Participants were recruited from a regular preschool in Hangzhou, China. This study was approved by the Research Ethics Board of Center for Psychological Sciences at Zhejiang University (No. 2020-003) in May 2020. Before measurements, a comprehensive verbal description of the nature and purpose of the study was given to the children and their teachers and parents before any measurements. A questionnaire was also sent to parents to identify any presence among participants of developmental problems or interest in cooperation, which were all considered exclusion criteria. Table 1 presents the descriptive characteristics of the preschool participants. They were grouped according to chronological age: 3-year-old group (*n* = 48), 4-year-old group (*n* = 70) and 5-year-old group (*n* = 76).

### 2.2. Procedures

The MSR test was administered using a specially constructed box with a slide ruler attached to the top. The height of the box was 26.5 cm. Before testing, the participant was seated on the box with head, back and hips against a wall (90° at hip joint), both legs together and fully extended, and the feet placed flat against the box. The participant then extended arms forward, placing the hand on the ruler with palms down (i.e., the starting point (P0)). Anthropometric characteristics were collected, including sitting height, arm length and base length in a starting position (Figure 1). Sitting height (SH) was vertically measured from acromion to the floor with head, back and hips against the wall. Arm length (AL) was measured from acromion to the middle fingertip with arm fully extended. Base length (BL), the approximate leg length, was measured parallel to the floor from the wall to the box with legs fully extended and feet placed against the box (90° at ankle joint). Then, the participant was asked to slowly reach forward as far away as possible, keeping the hips in contact with the wall, sliding the hands across the top of the ruler (i.e., the reaching point (P1)). Each participant performed the MSR test twice, and was allowed to rest 1–2 min between trials. All tests were administered by the same researcher. Test was repeated if the participant did not complete standard testing procedures (i.e., without legs fully extended or without hips in contact with the wall).

### 2.3. MSR Angle Calculation

Figure 1 shows the calculation of the MSR angle, which was proposed to estimate hamstring flexibility. It has been considered that the larger angle, the better the hamstring flexibility. MSR angle (°) was calculated as follows:L2=BL+P1
L1 = 26.5
$$\mathrm{Hip}\;\mathrm{flexion}\;\mathrm{angle}\;(\alpha )=\cos^{-1}(\frac{L1^{2}+SL^{2}+L2^{2}-AL^{2}}{2SL\sqrt{L2^{2}+L1^{2}}})+\cos^{-1}(\frac{L2}{\sqrt{L2^{2}+L1^{2}}})$$
MSR angle=90°−α
where L1 represents the height of box (i.e., 26.5 cm); SL represents the sitting height; AL represents the arm length; BL represents the base length (i.e., the length between the wall and the feet against the box); P1 represents the reaching point.

### 2.4. MSR Distance Calculation

Figure 2 shows the calculation of the MSR distance, according to the MSR procedures described by Hoeger et al. [5,6]. MSR distance (cm) was calculated as follows: MSR distance = P1 − P0 (cm).

### 2.5. Statistical Analyses

Statistical analysis was performed in SPSS software (v24.0 for Mac, SPSS Inc., Chicago, IL, USA), and the alpha level was set at *p* < 0.05. The normality test for data sets was down using the Shapiro–Wilk test (0.831–0.976, *p* > 0.05) and by visual observation of the normal Q–Q plots. MSR results were found to be normally distributed.

Coefficient of variation (CV%) was used to study the absolute reliability (i.e., within-subject variation) according to the following formula: typical error/mean value of the trials ×100%, where typical error (TE) was calculated by dividing the standard deviation of the trial-to-trial difference score by $\sqrt{2}$ [11]. Intraclass correlation coefficient (ICC) and its 95% confidence intervals (CI) were used to study the relative reliability. ICC (CI 95%) was calculated based on a mean-measurement (k = 2), absolute-agreement, 2-way mixed-effects model.

The usefulness of a test reflects the ease of identifying a change in performance [12]. Comparing TE with the smallest worthwhile change (SWC) was used to study the usefulness of two different MSR calculation methods. SWC_0.5_ was derived from the between-subject SD multiplied by 0.5 [13]. A TE below SWC indicated test usefulness to be “good” [12].

Then, a maximum of two trials on two different calculation methods was used for subsequent analyses. Pearson’s correlations were used to estimate the relation between anthropometric measurements and two different MSR results. The strength of the correlation was interpreted using the following qualitative descriptors: ≤0.20 = very weak, >0.20–0.40 = weak, >0.40–0.70 = moderate, >0.70–0.90 = strong and >0.90 = very strong [14].

Gender and age group comparisons of two different MSR results were performed by two-way ANOVA. Effect size was calculated using partial eta-squared. The post hoc tests using the LSD were done for different levels of each factor.

## 3. Results

Descriptive, reliability and usefulness data for the two MSR scores were presented in Table 2. The relative variability for MSR angle (ICC = 0.91 and 0.89) was shown to be better than for MSR distance (ICC = 0.86 and 0.84) in both 3- and 5-year-old groups. The absolute reliability for MSR angle (CV% ranging from 8.21 to 9.40) was shown to be better than that of MSR distance (CV% ranging from 9.29 to 10.63), with CV% values being smaller in three age groups. In the MSR angle, the TE exceeded SWC_0.5_ in 4-year-old group, and was smaller than SWC_0.5_ in 3- and 5-year-old groups. In the MSR distance, by contrast, the TE exceeded SWC_0.5_ in the 5–6-year-old group, and was smaller than SWC_0.5_ in the 3- and 4-year-old groups.

Pearson correlation coefficients were calculated for the two different MSR calculation results with height, sitting height and arm length to provide reliability estimates (Table 3). MSR distance and anthropometric characteristics (i.e., sitting height and arm length) were found to be weakly correlated (*p* < 0.05). In contrast, there was no significant correlation between MSR angle and anthropometric measurements.

Gender and age group characteristics of two different MSR results were presented in Table 4. Two-way ANOVA revealed a significant main effect of gender for both MSR distance scores (F (1.188) = 6.435, *p* = 0.012, η^2^ = 0.259) and MSR angle scores (F (1.188) = 5.000, *p* = 0.026, η^2^ = 0.143). LSD post hoc comparison reflected that girls aged 5 years received higher MSR angle scores than boys (*p* = 0.040).

## 4. Discussion

The present study proposed a method for calculating the approximate hip flexion angle to estimate the hamstring flexibility in preschoolers. Our results supported the good reliability of angular analysis for measuring hamstring flexibility during an MSR test. Besides, calculating the approximate hip flexion angle during an MSR test was able to eliminate the influence of anthropometric characteristics.

### 4.1. Reliability and Usefulness

Previous studies have reported the relative reliability of MSR reaching scores with acceptable ICC values of 0.84 in professional futsal players [15] and 0.97 in university students [16]. Findings from the current study are consistent with the previous studies, showing a good test-retest reproducibility in MSR distance scores in preschoolers (ICC values ranging from 0.84–0.91 in three age groups). Besides, a better test-retest reproducibility in the MSR angle scores was found than in the MSR distance scores among 3- and 5-year-old age groups. This means that the preschoolers maintained their ranking order more consistently relative to others in the group when performing the MSR test using angular analysis [17]. The absolute reliability was shown to be lower for MSR angle scores than MSR distance scores, reflected in the smaller CV% values in the MSR angle scores among preschoolers (Table 2).

As the units of the two MSR scores are different, the calculation of TE cannot be directly used as a method to evaluate absolute reliability. In this study, however, the usefulness is established by comparing the TE and SWC_0.5_ [12]. Our results showed that TE was smaller than SWC_0.5_ in 3-year-old and 5-year-old groups, suggesting good usefulness of the MSR angular analysis. That is, the MSR angle score can be utilized to detect moderate changes that exceed 0.5 times the test’s SD, showing good measurement usefulness in preschoolers. To summarize, the MSR angle scores showed good ability to detect and measure between-subject differences and within-subject differences in 3- and 5-year-old age groups.

### 4.2. Elimination of the Concern of Limb-Length Bias

A score on the SRT may be the result of a variety of factors, including pelvic position [4], scapular abduction [18], ankle dorsiflexion or plantarflexion [9], leg position (i.e., together or V-shaped) [19] and anthropometric characteristics [8]. Mainly, it has been reported that the finger-to-box distance (i.e., a relative zero point for each participant based on proportional differences in limb lengths) was correlated with the classical SR test scores., while there was no difference of MSR test scores among low, medium and high finger-to-box distance groups. For this reason, the MSR test was proposed. The present study, however, showed a weak correlation between MSR distance and anthropometric characteristics (Table 3). The results corroborated a previous study, in which the correlation between MST scores with arm span, arm length, leg length and trunk length were all significant (coefficient r ranging from 0.23 to 0.44) [8]. Therefore, researchers mainly aimed to eliminate the concern of disproportionate limb-length bias.

In practice, participants with long legs and short arms have a structural disadvantage when performing SR test, and are possible to receive lower scores than participants with short legs and long arms who achieve the same angle of hip flexion. Consequently, previous studies suggested the measurement of the hip joint angle (HJA), using the inclinometer or the angular kinematic analysis, might be more effective than classical SRT scores because the HJA scores are not influenced by anthropometric factors or spinal mobility. Previous studies found a strong correlation in school-aged children and a moderate correlation in young adults between the SRT and HJA measurements using the classical SR test, respectively [4,9] Similarly, we proposed the MSR angle for calculating the approximate hip flexion angle because flexibility is typically characterized by the maximum range of motion in a joint or series of joints [10]. In line with these previous studies, our present results showed a moderate correlation and no significant difference between the normalized MSR distance and MSR angle. Furthermore, our results also found that the correlation between MSR angle and anthropometric characteristics indicated no significant correlation (Table 3), which has not been reported in the previous studies.

### 4.3. Practical Implications

Due to the qualified technicians, additional devices and time constraints, the use of the angular tests seems to be limited in several settings including a school context. This study provides a precise method to sports coaches and physical education teachers about hamstring flexibility evaluation in preschoolers. The study found that the proposed MSR angular analysis can negate the effects of anthropometric characteristics without changing testing procedures and testing devices. Since children with Duchenne muscular dystrophy [20] and with a history of plagiocephaly [21] have been reported to have poor flexibility, it can be considered that such an MSR angular analysis might have good application prospects in children with typical or atypical development.

### 4.4. Limitations

From a practice perspective, this study has several potential limitations. First, we involved preschool-aged participants, which limits the external validity of the results. The future study might focus on the application of MSR angular analysis in other age groups, including school-aged children, young and older adults. Second, additional anthropometric measurements in the present study were obtained in a starting position using a tape measure, which might include some measurement errors. To minimize measurement errors, the further study might combine the digital measurement tools with the MSR angular analysis. For example, several displacement sensors can be placed on the sit-and reach device to automatically obtain the anthropometric measurement, including sitting height and base length.

## 5. Conclusions

In conclusion, the present results suggest that the proposed MSR angular analysis shows good absolute and relative reliability, and can eliminate the concern regarding the limb length bias or the device limitation. In practice, the use of the angular analysis seems to be accurate and feasible in a school context, since (a) the calculation of the proposed MSR angle when performing the MSR test requires a few additional anthropometric measurements, and (b) the testing procedure and the testing box remain unchanged.

## Figures and Tables

**Figure 1 children-09-00039-f001:**
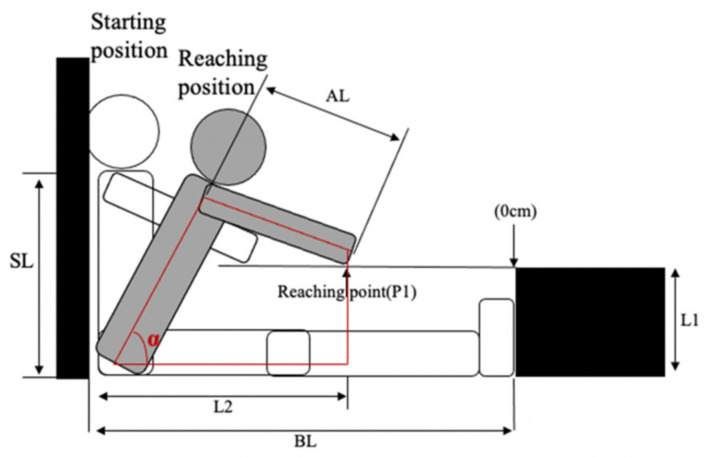
Illustration for MSR angle. SL, sitting height; BL, base length; AL, arm length; the height of box is 26.5 cm. An example of a 4-year-old boy: SL = 31 cm; BL = 70 cm; AL = 44 cm; P1 = −5 cm. Then his MSR angle score is 43.07°.

**Figure 2 children-09-00039-f002:**
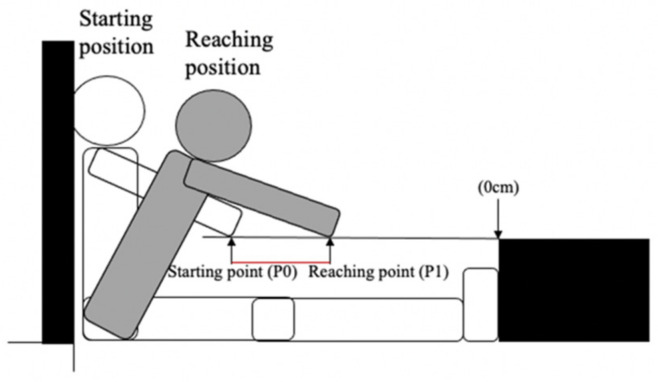
Illustration for MSR distance. An example of a 4-year-old boy: starting point is −16 cm and reaching point is −5 cm. Then his MSR distance score is 11 cm.

**Table 1 children-09-00039-t001:** Descriptive characteristics of the participants.

Age Group	N	Gender (M/F)	Height (cm)	Weight (kg)
3 years	48	28/20	102.99 ± 4.73	16.48 ± 2.48
4 years	70	44/26	109.40 ± 4.67	18.48 ± 3.09
5 years	76	47/29	114.49 ± 4.53	20.02 ± 2.67

**Table 2 children-09-00039-t002:** Descriptive and reliability parameters for MSR angle and MSR distance scores in three age groups.

	3 Years					4 Years					5 Years				
	Mean (SD)	ICC (95% CI)	CV%	TE	SWC_0.5_	Mean (SD)	ICC (95% CI)	CV%	TE	SWC_0.5_	Mean (SD)	ICC	CV%	TE	SWC_0.5_
Angle (°)	46.29 (10.33)	0.91 (0.79–0.95)	8.21	3.80	5.17	48.52 (8.45)	0.82 (0.70–0.89)	9.23	4.48	4.22	47.35 (10.89)	0.89 (0.82–0.93)	9.40	4.45	5.44
Trial 1	44.90 (10.03)					47.52 (8.27)					46.16 (11.25)				
Trial 2	47.69 (10.54)					49.52 (8.56)					48.54 (10.44)				
Distance (cm)	19.05 (3.96)	0.86 (0.69–0.93)	9.29	1.77	1.98	20.65 (4.94)	0.91 (0.86–0.94)	9.30	1.92	2.47	21.44 (4.37)	0.84 (0.75–0.89)	10.63	2.28	2.18
Trial 1	18.85 (4.22)					20.30 (4.96)					21.20 (4.44)				
Trial 2	20.15 (3.61)					21.01 (4.94)					21.69 (4.31)				

ICC = intraclass correlation coefficient; CV% = coefficient of variation; TE = typical error; SWC = smallest worthwhile change; 95% CI, 95% confidence interval.

**Table 3 children-09-00039-t003:** Intercorrelations of the MSR angle, the MSR distance, height, sitting height and arm length in all participants (*n* = 194).

	MSR Angle	MSR Distance	Height	Arm Length	Sitting Height
MSR angle		0.430 ***	0.041	−0.005	−50.006
MSR distance			0.127	0.209 **	0.339 ***
Height				0.841 ***	0.637 ***
Arm length					0.719 ***

*** Correlation is significant at the 0.001 level (two tailed). ** Correlation is significant at the 0.01 level (two tailed).

**Table 4 children-09-00039-t004:** Two-way ANOVA results of MSR angle scores and MSR distance scores for age and gender groups.

Variables	3 Years		4 Years		5 Years	
	Boys	Girls	Boys	Girls	Boys	Girls
Angle (°)	46.37 (10.12)	46.19 (9.41)	47.39 (7.64)	50.42 (7.53)	45.28 (11.69)	50.72 (6.28) *
Distance (cm)	18.90 (2.96)	20.33 (4.37)	20.12 (4.79)	21.54 (4.47)	20.62 (3.92)	22.78 (3.87)

* Significant difference between girls and boys in 5–6-year-old group, *p* < 0.05.

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
