# Peer review of "Comparison of Distance and Angular Analysis for Measurement of Hamstring Flexibility in Preschoolers"

_children, 2022, doi:10.3390/children9010039_

Round 1

Reviewer 1 Report

The present manuscript shows that the angular analysis when performing the MSR test may eliminate the concern regarding the limb length bias. It describes interesting findings for Children readers, but could be improved:

a) Please, do not include stats in the abstract, and add some implication for practice.

b) Further study justification is recommended, concerning what is the gap in the literature and its relevance. Hypotheses may be also further described and justified with previous research.

c) More information regarding socioeconomic status, educational level of the family and habitat, could be described for the sample.

d) In the results, effects sizes should be reported.

e) Normality test is recommended before selecting the statistics.

f) Practical implications may be developed in the discussion. As well, limitations and future research lines in this area need to be further described.

Reviewer 2 Report

Overall comments - The manuscript is well-written with clear findings and potential implications for the field.

General comments

The introduction flows nicely and logically through prior literature followed by the identification of the problem and purpose of the study, potential practical applications, and the hypotheses.

The intent of the study has practical applications to the field in preschool settings.

The figures are supportive of the description of the procedures.

How were children broken into age groups when the ages appear to overlap (3-4 years, 4-5 years, 5-6 years)?

Tables supported the textual representation of the results.

The discussion section included clarification and emphasis on key findings.

Good usefulness of angular analysis was found in the 3-4-year-old and 5-6-year-old age groups based on comparisons of typical error and smallest worthwhile change.  Typical error was greater than smallest worthwhile change in the 4-5-year-old group, but this was not addressed, and “good measurement usefulness in preschoolers” was declared.  The authors may wish to address this gap.

Within the limitations section, the first and fourth sentences pertain to age while the second and third sentences focus on gender-related findings and applications.  Re-organizing this paragraph is advised.

The conclusion closes the loop on a key point in the introduction regarding feasibility in a preschool setting.

Line-by-Line comments

11 – spell out modified sit-and-reach first time in the abstract and introduce the acronym

53 – should be ”series of joints” rather than “series joint”

120 – should be “his” rather than “hos”
